

# Biomechanical evidence suggests extensive eggshell thinning during incubation in the Sanagasta titanosaur dinosaurs

E. Martín Hechenleitner[1], Jeremías R. A. Taborda[2], Lucas E. Fiorelli[1], Gerald Grellet-Tinner[1,3] and Segundo R. Nuñez-Campero[1]

[1] Centro Regional de Investigaciones Científicas y Transferencia Tecnológica de La Rioja (CRILAR), Provincia de La Rioja, UNLAR, SEGEMAR, UNCa, CONICET, Anillaco, La Rioja, Argentina
[2] Centro de Investigaciones en Ciencias de la Tierra (CICTERRA), Universidad Nacional de Córdoba, CONICET, FCEFyN), Córdoba, Argentina
[3] The Orcas Island Historical Museums, Eastsound, WA, USA

## ABSTRACT

The reproduction of titanosaur dinosaurs is still a complex and debated topic. Their Late Cretaceous nesting sites are distributed worldwide and their eggs display substantial morphological variations according to the parent species. In contrast to the typical 1.3–2.0 mm thick shells common to eggs of most titanosaur species (e.g., those that nested in Auca Mahuevo, Tama, Toteşti or Boseong), the Cretaceous Sanagasta eggs of Argentina display an unusual shell thickness of up to 7.9 mm. Their oviposition was synchronous with a palaeogeothermal process, leading to the hypothesis that their extra thick eggshell was an adaptation to this particular nesting environment. Although this hypothesis has already been supported indirectly through several investigations, the mechanical implications of developing such thick shells and how this might have affected the success of hatching remains untested. Finite element analyses estimate that the breaking point of the thick-shelled Sanagasta eggs is 14–45 times higher than for other smaller and equally sized titanosaur eggs. The considerable energetic disadvantage for piping through these thick eggshells suggests that their dissolution during incubation would have been paramount for a successful hatching.

## INTRODUCTION

Recent studies have changed our perspective on titanosaur palaeobiology. These highly diversified dinosaurs were the largest terrestrial organisms that ever roamed the earth and, according to recent investigations, their thermophysiology was similar to that of large modern endotherms (*Seymour et al., 2012*; *Seymour, 2013*; *Eagle et al., 2015*). Titanosaur eggs were incubated in holes excavated in the soil or in mounds of soil and leaf litter, comparable to the nests of the modern megapodes (*Grellet-Tinner & Fiorelli, 2010*; *Hechenleitner, Grellet-Tinner & Fiorelli, 2015*) and their chicks had a rapid ontogenetic development (*Werner & Griebeler, 2014*; *Curry Rogers et al., 2016*). Perinatal embryos preserved in ovo also revealed that titanosaurs developed an "egg-tooth"-like

Corresponding author
E. Martín Hechenleitner,
emhechenleitner@gmail.com

**Table 1  Avian and non-avian dinosaur eggs used in the comparative analyses.**

| Living birds | Thickness [mm] | V [L] | E [Gpa] | Load point | | | Source |
|---|---|---|---|---|---|---|---|
| | | | | X1 | X2 | X3 | |
| Quail | 0.22 | 0.00484732 | 10.5 | 5.73642 | −9.61573 | −9.75E-14 | *Hahn et al. (2017)* |
| Hen | 0.41 | 0.028573099 | 18 | 10.8715 | −17.8913 | 6.46E-14 | |
| Goose | 0.67 | 0.064067514 | 10.4 | 14.725 | −24.2459 | 1.09E-13 | |
| Ostrich | 2.55 | 0.456017893 | 6.6 | 26.6064 | −46.141 | 2.37E-14 | |
| **Titanosaur** | **Thickness [mm]** | **V [L]** | **d [mm]** | **Load point** | | | **Source** |
| | | | | X1 | X2 | X3 | |
| Tama | 1.495 | 2.50516882 | 167.01 | 41.75 | 72.3131 | 3.09E-09 | *Hechenleitner et al. (2016b)* |
| Boseong | 1.765 | 2.639751625 | 171.47 | 42.8675 | 74.2487 | −6.77E-15 | *Huh & Zelenitsky (2002)* |
| Auca Mahuevo | 1.39 | 1.352853804 | 137.22 | 34.3 | 59.4093 | −1.62E-14 | *Grellet-Tinner, Chiappe & Coria (2004)* |
| Toteşti | 1.75 | 1.19220506 | 126.5 | 31.625 | 54.7761 | 2.62E-14 | *Grellet-Tinner et al. (2012)* |
| Sanagasta | 1.2–7.95 | 2.53576055 | 169.188 | 42.297 | 73.2606 | 5.49E-14 | *Grellet-Tinner & Fiorelli (2010)* |

Notes:
Specifications for each egg model. d, inner diameter. E, Young's modulus (for all titanosaur models this value is 17.51 GPa). V, inner volume. X1, X2, X3, spatial coordinates of the load point.

structure (*García, 2007*) that could have served to break the shell during hatching. Such anatomical structure is present in all the archosaurs (from crocodilians to birds) and presently, is the only known to be specifically involved in the hatching process (*Honza et al., 2001*; *García, 2007*; *Hieronymus & Witmer, 2010*; *Hermyt et al., 2017*).

Titanosaurs laid amniotic eggs with a calcitic shell. This genetically and physiologically controlled, biomineralized hard layer that protects the developing embryo from damage (mechanical or chemical), dehydration and infection, is specifically adapted to particular nesting environments, hence functionally optimized for each species (*Ferguson, 1981*; *Board, 1982*). Titanosaur eggshells consist of monolayered calcium carbonate, growing from densely packed shell units of rhombohedric, acicular calcite crystals that radiate from nucleation centers located at the external surface of the membrana testacea (*Grellet-Tinner, Chiappe & Coria, 2004*). Although titanosaur eggshells typically are 1.35–2.0 mm thick, the exceptionally thick-shelled eggs of the Sanagasta nesting site, in La Rioja, Argentina, reach 7.9 mm (*Grellet-Tinner & Fiorelli, 2010*; *Grellet-Tinner, Fiorelli & Salvador, 2012*; *Hechenleitner et al., 2016a*) (Table 1).

At Sanagasta, more than 80 titanosaur egg clutches were found to be synchronous with a Cretaceous geothermal process (*Grellet-Tinner & Fiorelli, 2010*; *Fiorelli et al., 2012*). Although unique among non-avian dinosaurs, the evidence at hand suggests that several species of titanosaurs may have utilized geothermalism as a source of heat for egg incubation (*Grellet-Tinner & Fiorelli, 2010*; *Hechenleitner, Grellet-Tinner & Fiorelli, 2015*). Yet, nesting in active geothermal settings is still a strategy exploited by several modern

vertebrates, chiefly iguanas, snakes, birds, and even deep-sea skates (*Werner, 1983*; *Göth & Vogel, 1997*; *Guo et al., 2008*; *Salinas-de-León et al., 2018*), because it ensures a nesting thermal stability. Such association between titanosaur nesting and palaeogeothermalism led to hypotheses that thickness of the Sanagasta eggshells was an adaptation to resist the extrinsic dissolution by pore fluids in a harsh nesting environment (*Grellet-Tinner & Fiorelli, 2010*; *Grellet-Tinner, Fiorelli & Salvador, 2012*). This hypothesis received additional paleobiological support from more recent studies on the striking thickness of these eggshells (*Grellet-Tinner, Fiorelli & Salvador, 2012*; *Hechenleitner et al., 2016a*). The new data confirmed that these titanosaur eggs were physiologically functional; that is, they would have allowed an appropriate gas exchange under burial conditions in the substrate, even when their shells were as thick as 7.9 mm. Moreover, calculations based on micro-CT data showed that the eggshells were also physiologically functional even when they thinned up to 80% or 1.5 mm (*Hechenleitner et al., 2016a*). This implies that the suggested external chemical erosion of the shell by hydrothermal fluids would not have compromised the incubation with respect to gas exchange. However, whether or not this dissolution of the shell was essential for the hatchability of the Sanagasta eggs (as well as other titanosaur eggs) is a hypothesis that has not yet been tested.

Therefore, the present investigation aims to test the mechanical strength of the Sanagasta eggs using finite element analyses (FEA) on models of titanosaur eggs from several nesting sites by evaluating the required force to break them from inside. Furthermore, it will shed light on the importance of the external dissolution of the shell by chemical leaching, and its paramount role for their hatchability and the survival of several titanosaur species.

## METHODS

### Specimens and modeling

We analyzed data of Haţeg (Romania), Boseong (South Korea), Tama, Sanagasta, and Auca Mahuevo nesting sites (Argentina) (Table 1). Measurements of the eggs from Tama and Sanagasta were obtained from digital 3D reconstructions of specimens curated at the Centro Regional de Investigaciones Científicas y Transferencia Tecnológica de La Rioja (CRILAR-PV 530/1 and CRILAR-PV 400 SA-C6-e1, respectively). Egg models for other sites are based on personal observations (Haţeg and Auca Mahuevo) and literature (Boseong) (*Hechenleitner, Grellet-Tinner & Fiorelli, 2015*; *Hechenleitner et al., 2016b*). In addition, we included data from *Hahn et al. (2017)* for four kinds of living birds: quail, hen, goose, and ostrich (Table 1). A comparison of their size and shape is given in Fig. 1A.

### Egg morphology and size

In most nesting sites the titanosaur eggs are transformed, mostly compressed, during diagenesis; hence, it is difficult to assess exactly their original shape and diameter (*Hechenleitner et al., 2016b*). Therefore, we performed a CT-scan of a complete egg from Sanagasta (CRILAR-Pv 400 SA-C6-e1), using a 64-channel multi-slicer tomograph, at 140 Kv and 403 mA. The resulting CT dataset was analyzed by using 3D Slicer v4.1.1 (*Fedorov et al., 2012*) and we obtained 141 three-dimensional structures that correspond

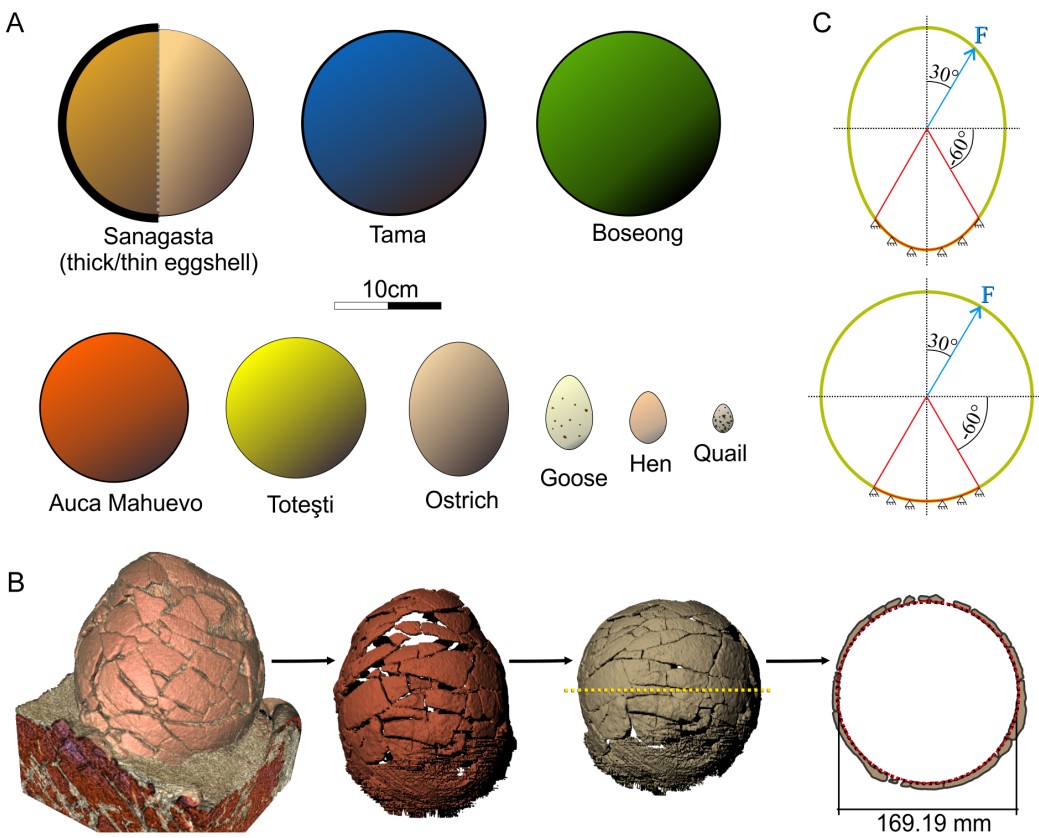

**Figure 1 Dinosaur eggs.** (A) Schematic silhouettes of the titanosaur and modern bird eggs used in the mechanical analyses. (B) Reconstruction of CRILAR-Pv 400 SA-C6-e1. (C) Boundary conditions for the analyses. Study sites: F, inner load force.

to eggshell fragments. During the analysis of the CT we observed that the ellipsoidal shape of the egg CRILAR-Pv 400 SA-C6-e1 is a product of the displacement of shell fragments by the sediment. Using CAD software (DesignSpark Mechanical v.2015.0), we relocated each fragment to its original relative position (Fig. 1B). This produced an assembled model of spherical shape. Using this model we estimated the inner volume (2,500 cm$^3$) and inner diameter (169 mm), required to make the finite element model (FEM).

Size estimations of the eggs from Toteşti and Tama are based on CT data (*Grellet-Tinner et al., 2012*; *Hechenleitner et al., 2016b*). The estimations for the eggs from Boseong and Auca Mahuevo (*Grellet-Tinner, Chiappe & Coria, 2004*; *Hechenleitner, Grellet-Tinner & Fiorelli, 2015*) should be taken with caution until CT scans provide accurate data. All measurements are summarized in the Table 1.

## Eggshell mechanical properties

The eggshells, like bones, loose their original mechanical properties during fossilization, hence biomechanical analyses must rely on data from living relatives. The titanosaur eggshells are homologous to the internal-most layer (layer 1 or mammillary layer) of the bird's eggshell (*Grellet-Tinner, Chiappe & Coria, 2004*). Recent insightful information with respect to the mechanical properties of the eggs of several living species of

birds (*Hahn et al., 2017*) allow overcoming of the limitations imposed by diagenesis for conducting FEA on titanosaur eggs. Input data for carrying out FEA was obtained from the empirical tests performed on birds' eggshells (*Hahn et al., 2017*). We selected average values from existing data (Table 1) for the calculations on titanosaur egg models. These are: Young's modulus (E) = 17.51 GPa and assumed a Poisson's ratio ($\nu$) = 0.3.

The shell of the amniote egg has a tremendous structural complexity, including organic and inorganic compounds (*Board, 1982*; *Bain, 1992*; *Juang et al., 2017*; *Hahn et al., 2017*) as well as voids (e.g., pore canals and vesicles). Because data was obtained through empirical tests (*Hahn et al., 2017*), measured mechanical properties result from the interaction of all of these variables. Hence, all the eggs were modeled using a homogeneous "eggshell" material with the mechanical properties of a modern bird's eggshell.

## Finite element models

The shape of the bird eggs varies considerably. As such, to construct the 3D egg models, we used the outline of the eggs shown by *Hahn et al. (2017)* and assume each egg as a revolved solid. The titanosaur eggs were modeled following the same protocol, although, based on previous data (*Hechenleitner et al., 2016b*), we assumed a 2D circular outline. Thickness of the revolved solids in all cases is equivalent to that of the respective eggshell. All models were made using CAD software (Fig. 1A).

To define the boundary conditions of the FEMs we located the center of the egg in the middle of its maximum-length axis (Fig. 1C). The external surface was fixed below 150°, to avoid rotation of the models.

In contrast to external resistance tests found in the literature (*Juang et al., 2017*; *Hahn et al., 2017*), in which a force is applied on the apex of the eggs, we decided to apply the internal force in an angle similar to that observed in birds during hatching. In modern birds the hatching point is variable, between the equator and the blunt end of the egg. As such we selected a 30° angle from the maximum-length axis to apply the load force. The latter angle is only important for the asymmetric eggs, because the shell does not mechanically behave uniformly.

In the present work we evaluate the structural response of the eggs to an internal force, emulating the conditions of effort during hatching. Because the egg is a closed structure, it is impossible to do such empirical tests without damaging the shell. In a recent paper, *Juang et al. (2017)* show that the eggs of all avian species fractured from outside at a displacement to thickness ratio of about 1. Because of its shape, the structural behavior of the egg is different from the internal and external side. However, although the actual ratio may vary, the ratio = 1 was used as a simplified criterion to determine the fracture force. This means that we assumed that the shell breaks when the displacement at the load point equals its thickness. As such, our model seeks to obtain a parameter in equivalent conditions among different eggs, which allows comparison of the mechanical performance during hatching.

All models were meshed using tetrahedral elements of four nodes (see supplementary *.nas files), considering that the eggshell material is isotropic and homogeneous. The

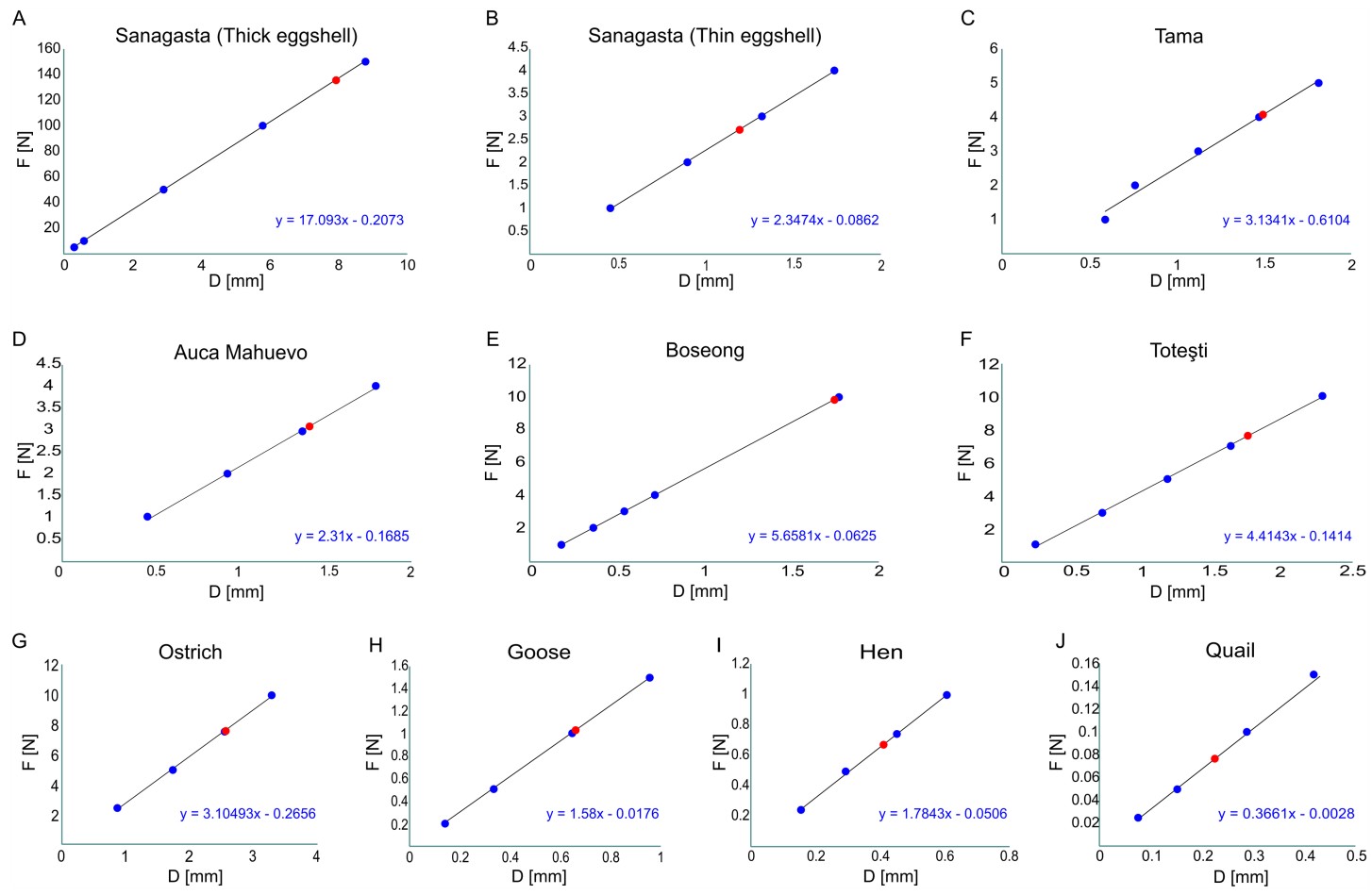

**Figure 2 Break point estimations for each egg model.** (A) Sanagasta eggs with the thickest shell reported for this site. (B) Sanagasta eggs with the thinnest shell reported for this site. (C) Tama. (D) Auca Mahuevo. (E) Boseong. (F) Toteşti. (G) Ostrich. (H) Goose. (I) Hen. (J) Quail. Blue dots, FEA results for each test. Red dot, break point estimated by the regression. Results are given in Table 2.

elastic properties of each egg model are specified in Table 1. The finite element analyses were conducted using the software ADINA v8.7.3.

## Breaking force estimation

In all instances (birds and titanosaurs), we conducted exploratory analyses. Using internal forces of different magnitude we recorded the eggshell displacement at the load point (Figs. 2A–2J; Table 2). Based on these results, we estimated the inner load force required to obtain a displacement equal to the eggshell thickness in each case (Fig. 3; Table 2).

## Effect of the eggshell dissolution on the egg mechanical resistance

In order to evaluate the effect of the dissolution of the eggshell in the Sanagasta eggs, as was previously hypothesized (*Grellet-Tinner & Fiorelli, 2010*; *Grellet-Tinner et al., 2012*; *Hechenleitner et al., 2016b*), we generated and analyzed models with different shell thicknesses between 7.9 and 1.2 mm (the maximum and minimum thicknesses

**Table 2 Summary of the breaking force tests for each egg model.**

| Model | T# | F [N] | D [mm] |
|---|---|---|---|
| **Tama** | T1 | 1 | 0.5910 |
| | T2 | 2 | 0.7619 |
| | T3 | 3 | 1.1230 |
| | T4 | 4 | 1.4720 |
| | T5 | 5 | 1.8120 |
| | BP | 4.077752 | 1.4950 |
| **Sanagasta** (thick eggshell) | T1 | 5 | 0.2934 |
| | T2 | 10 | 0.5857 |
| | T3 | 50 | 2.8980 |
| | T4 | 100 | 5.7960 |
| | T5 | 150 | 8.7950 |
| | BP | 136.09665 | 7.9500 |
| **Sanagasta** (thin eggshell) | T1 | 1 | 0.4560 |
| | T2 | 2 | 0.8959 |
| | T3 | 3 | 1.3210 |
| | T4 | 4 | 1.7340 |
| | BP | 2.73 | 1.2000 |
| **Auca Mahuevo** (egg levels 1-3) | T1 | 1 | 0.4935 |
| | T2 | 2 | 0.9523 |
| | T3 | 3 | 1.3830 |
| | T4 | 4 | 1.7920 |
| | BP | 3.0424 | 1.3900 |
| **Toteşti** | T1 | 1 | 0.2400 |
| | T2 | 3 | 0.7200 |
| | T3 | 5 | 1.1800 |
| | T4 | 7 | 1.6300 |
| | T5 | 10 | 2.2800 |
| | BP | 7.5836250 | 1.7500 |
| **Boseong** | T1 | 1 | 0.1825 |
| | T2 | 2 | 0.3639 |
| | T3 | 3 | 0.5442 |
| | T4 | 4 | 0.7234 |
| | T5 | 10 | 1.7760 |
| | BP | 9.9240465 | 1.7650 |
| **Ostrich** | T1 | 5 | 1.7410 |
| | T2 | 10 | 3.3090 |
| | T3 | 2.5 | 0.8963 |
| | T4 | 7.5 | 2.5420 |
| | BP | 7.580065 | 2.5500 |

| Table 2 (continued). | | | |
|---|---|---|---|
| **Model** | **T#** | **F [N]** | **D [mm]** |
| **Goose** | T1 | 0.5 | 0.3736 |
| | T2 | 0.2 | 0.1517 |
| | T3 | 1 | 0.7295 |
| | T4 | 1.2 | 0.8675 |
| | BP | 0.919825 | 0.6700 |
| **Hen** | T1 | 0.25 | 0.1575 |
| | T2 | 0.5 | 0.3114 |
| | T3 | 0.75 | 0.4621 |
| | T4 | 1 | 0.6096 |
| | BP | 0.666208 | 0.4100 |
| **Quail** | T1 | 0.05 | 0.1515 |
| | T2 | 0.1 | 0.2946 |
| | T3 | 0.15 | 0.4306 |
| | T4 | 0.025 | 0.0769 |
| | BP | 0.07477 | 0.2200 |

**Note:**
BP, break point estimated by regression; D, maximum displacement at the load point; F, inner load force; T#, test number.

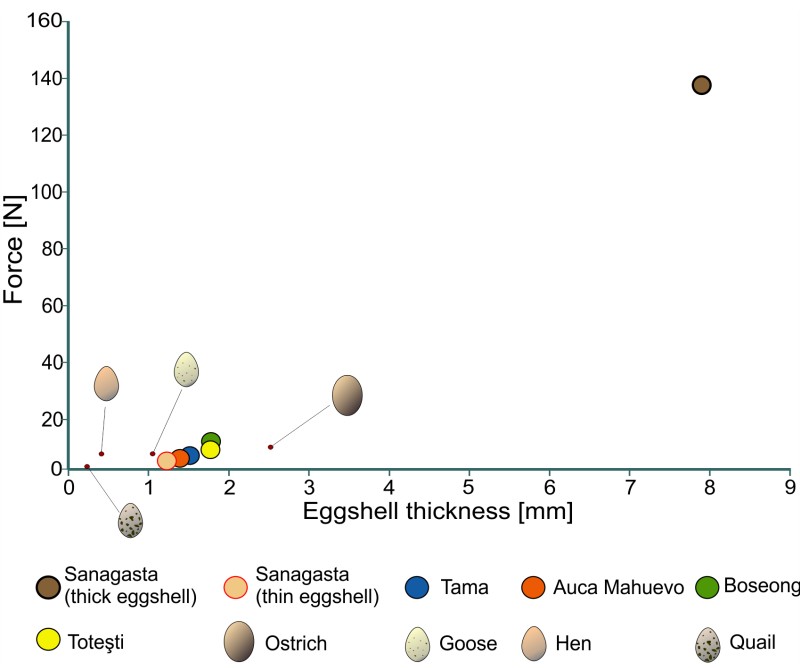

**Figure 3 Egg strength of several dinosaur eggs.** Fracture limit of each egg as a function of its shell thickness.

recorded at this site). Each of these models was evaluated with an internal load force of 5 N (Figs. 4A and 4B). This magnitude corresponds to the average of forces previously estimated for all the titanosaur eggs in our sample, excluding the estimation for maximum

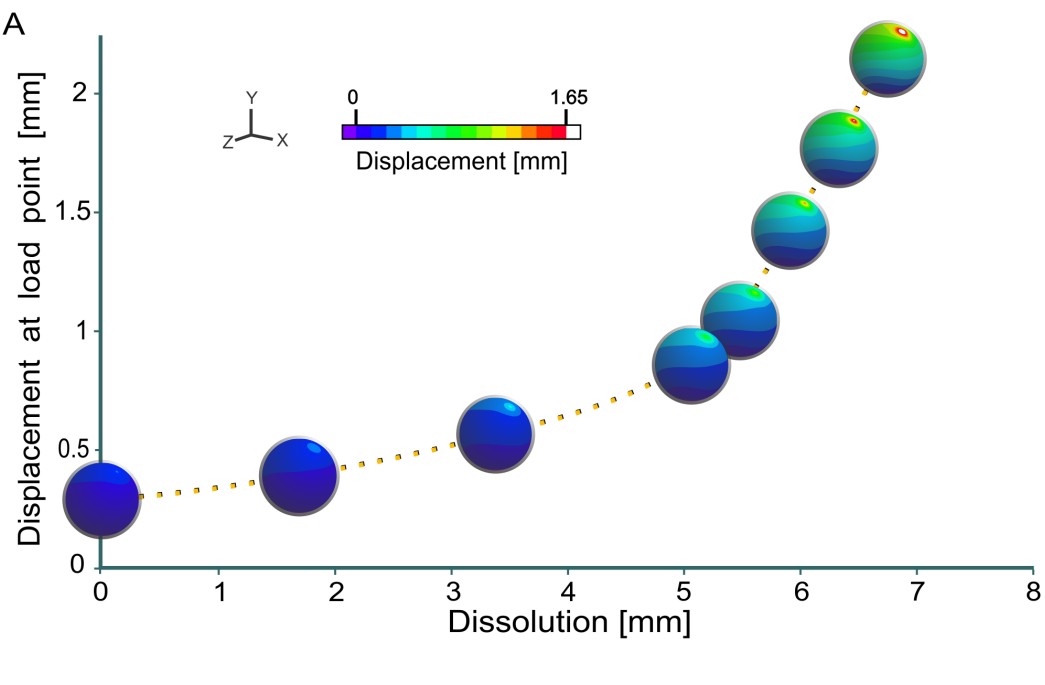

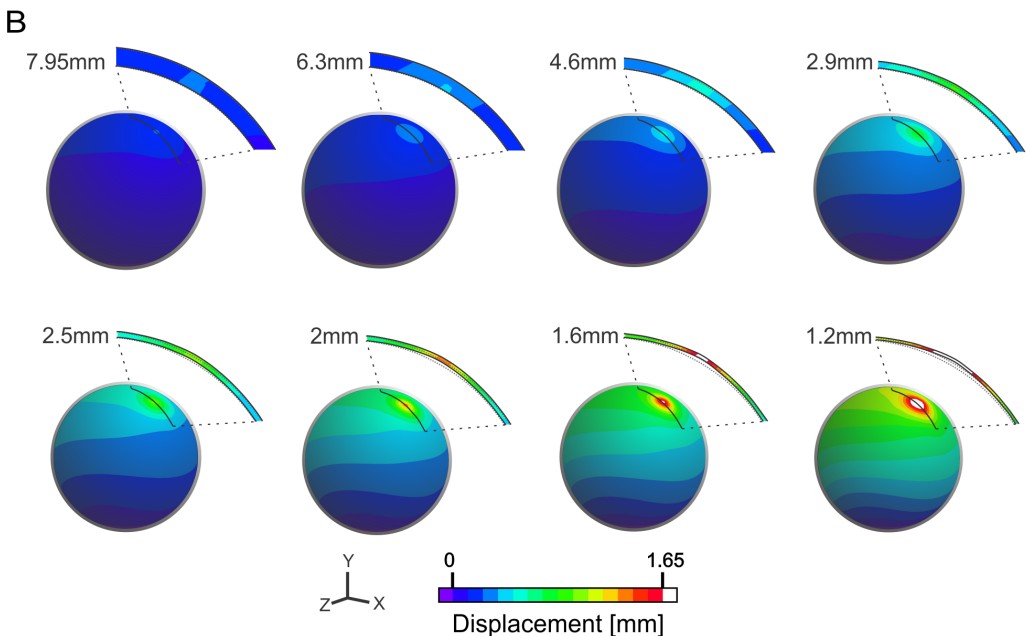

**Figure 4 Strength variations of the Sanagasta eggs.** (A) Strength variations as incubation progresses, according to *Grellet-Tinner & Fiorelli (2010)*. (B) Detail of strength variation for the Sanagasta eggs as thinning progresses. Note that displacement equals shell thickness when dissolution reaches ~6.3 mm (shell thickness = ~1.6 mm).

thickness of the Sanagasta eggs. Based on the data of maximum displacement at the load point (Table 3), we estimated the maximum shell thickness that can be broken applying 5 N.

**Table 3 Results of FEA on Sanagasta egg models with different eggshell thicknesses.**

| Eggshell thickness [mm] | Dissolution [mm] | Displacement at the load point [mm] |
|---|---|---|
| 7.95 | 0 | 0.29 |
| 6.26 | 1.69 | 0.39 |
| 4.58 | 3.38 | 0.56 |
| 2.89 | 5.06 | 0.87 |
| 2.47 | 5.48 | 1.05 |
| 2.04 | 5.91 | 1.42 |
| 1.62 | 6.33 | 1.78 |
| 1.2 | 6.75 | 2.14 |

## Statistical analysis

We performed a multiple linear regression analysis to test the influence of the egg volume and shell thickness on the strength of the eggs (Fig. 5). To perform the statistical analysis we used the lm function from the package stats version 3.4.3 of the open source software R (*R Development Core Team, 2017*).

Two models were performed in order to evaluate the relationship between variables; one model with interaction of the variables volume and thickness and one without interaction. The Akaike information criterion (AIC) method was used to select the model that better fits to data. A residual *vs* leverage plot of the fittest model helped to identify extreme values within the data set.

## RESULTS

According to the present 3D reconstruction, the Sanagasta eggs were originally spherical (Fig. 1B). This is consistent and supports all previous publications on titanosaur eggs (*Grellet-Tinner, Chiappe & Coria, 2004*; *Grellet-Tinner, Fiorelli & Salvador, 2012*; *Hechenleitner, Grellet-Tinner & Fiorelli, 2015*). Furthermore, the present CT-scan-based analysis shows that previous studies overestimated the size of these eggs (Fig. 1B). After digitally rearranging the eggshell fragments, the external egg diameter decreased from ~210 mm (~4,850 cm$^3$ in volume) to ~180 mm (~3,370 cm$^3$). Such a reduction in volume involves much less internal space for nutrient storage and embryo development. In addition, the diameter of the embryonic chamber of the Sanagasta eggs only reaches 169.2 mm due to the considerable shell thickness of these eggs (Fig. 1B). Therefore, although the Sanagasta eggs are larger than those of Tama, a nesting site found less than 150 km away in the same stratigraphic unit (*Hechenleitner et al., 2016b*), both display an identical chamber space available for the developing embryo (Table 1).

The 3D FEA conducted here, which are the first of their kind, allowed estimations that an effort of 3.04–9.77 N could break most of the titanosaur egg samples, namely Tama, Toteşti, Boseong, and Auca Mahuevo (Figs. 2A–2F and 3). In contrast, the eggs of Sanagasta are 14–45 times stronger, requiring up to 136 N to break.

Porosity could affect the eggshell's strength, although to date, there is no quantitative information in this regard (*Hahn et al., 2017*). Eggshell strength in modern birds has

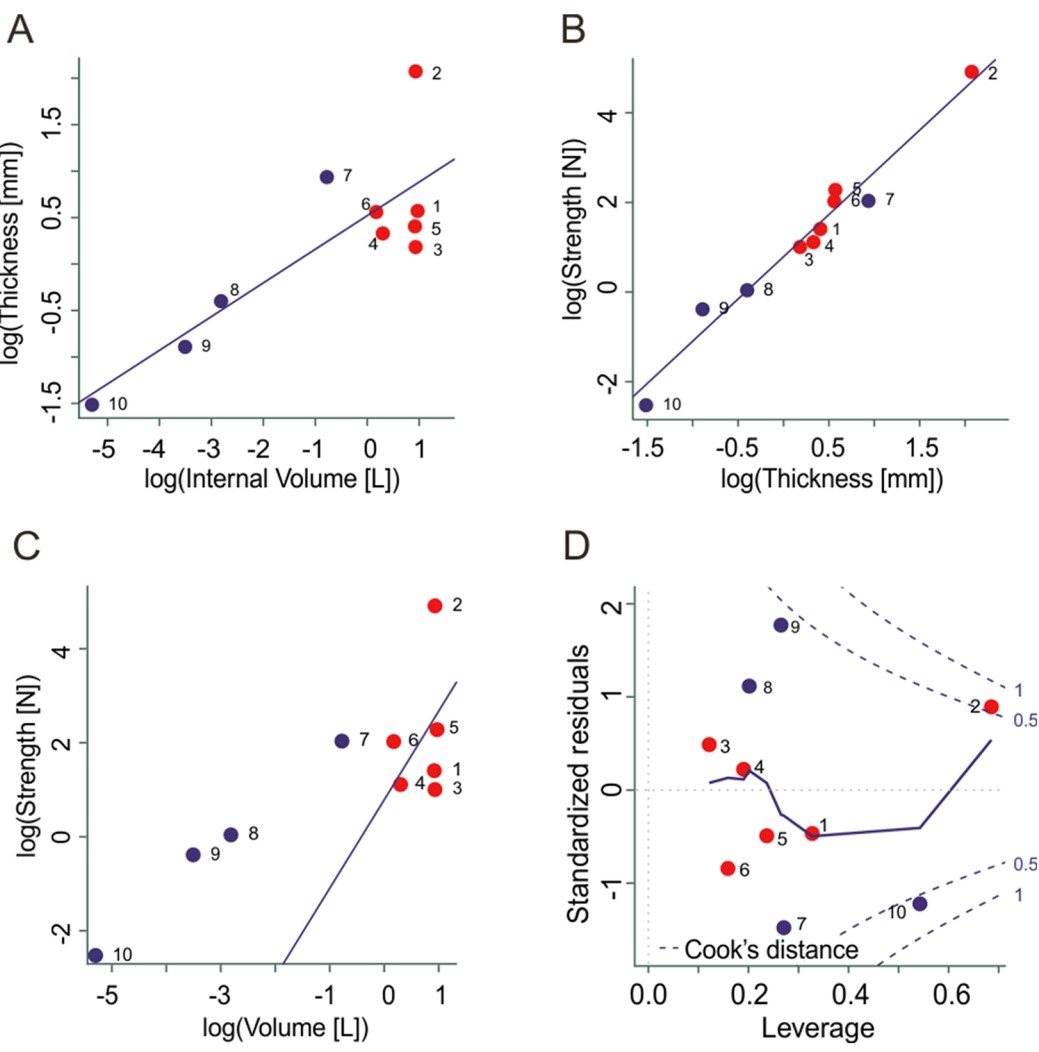

**Figure 5 Statistical analysis.** Multiple linear regression between: (A) Egg volume and shell thickness, (B) egg thickness and strength, and (C) egg volume and strength. (D) Model diagnostic plot of standardized residuals *vs.* leverage, showing the most extreme and influencing thickness values on the eggshell strength, corresponding to the thick-shelled eggs from Sanagasta (2) and the quail eggs (10). Red and blue dots correspond to titanosaur and avian eggs respectively. Reference numbers: (1) Sanagasta (thick); (2) Sanagasta (thin); (3) Tama; (4) Auca Mahuevo; (5) Boseong; (6) Toteşti; (7) Ostrich; (8) Goose; (9) Hen; (10) Quail.

been correlated with several factors, e.g., calcium diet, shell microstructure, incubation period; however, shell thickness is the main factor affecting strength (*Ar, Rahn & Paganelli, 1979*). The statistical model corroborated that there is an important linear association between egg internal volume and shell thickness ($F_{(1,8)} = 16.93$, $R^2 = 0.64$, $p = 3.40^{-4}$), although an over-dispersion of thickness values becomes evident as volume increases (Fig. 5A). From the two multiple linear regression models tested, the model that better explains the relationship between internal volume and eggshell thickness as independent variables, and the shell mechanical strength as response variable was the model without interaction (AIC = 14.13). The regression analysis showed a statistical association between eggshell thickness and the mechanical strength of the eggs ($F_{(2,7)} = 107.1$,

$R^2 = 0.96$, $p = 8.53^{-5}$; Fig. 5B), whereas there is not a direct association with egg internal volume ($F_{(2,7)} = 107.1$, $R^2 = 0.96$, $p = 0.80$; Fig. 5C). The residual *vs* leverage plot shows that the thick-shelled egg from Sanagasta and the quail egg represent outlier values, and according to the Cook's distance, they are strong influential observations for the model (Fig. 5D).

Considering that the geological and palaeontological data, as well as the evidence from modern analogues, suggest that the eggshells of Sanagasta would have partially dissolved during incubation, we further tested the mechanical effect of their constant thinning (Figs. 4A and 4B; Table 3). Results indicate that the average estimate for the other titanosaur eggs (5 N), has little effect on the Sanagasta egg, when its shell is thick (Figs. 4A and 4B). However, as the thinning progresses, the shell strength drops abruptly. When thinning reaches ~1.6 mm, the shell reaches its fracture threshold and, as previously speculated (*Grellet-Tinner, Fiorelli & Salvador, 2012*; *Hechenleitner et al., 2016a*), it breaks easily at and below this threshold (Figs. 4A and 4B).

## DISCUSSION

The concept that all of the eggs of titanosaurs are spherical is well established. However, several sites preserve deformed and/or incomplete eggs (*Huh & Zelenitsky, 2002*; *Salgado et al., 2009*; *Jackson, Schmitt & Oser, 2013*; *Hechenleitner et al., 2016b*), and there is little CT information available to reconstruct their original shape and volume. The CT scan of the specimen CRILAR-Pv 400 SA-C6-e1 confirmed that the Sanagasta eggs were spherical. A spherical shape in eggs is mechanically and physiologically optimal. It has a greater resistance to impacts and is the smallest surface with respect to any geometric figure of equal volume (*Bain, 1992*; *Stoddard et al., 2017*). As such it is advantageous in terms of strength, shell economy, and heat conservation (*Kratochvil & Frynta, 2006*; *Stoddard et al., 2017*).

Currently, there is strong evidence for titanosaurs' precociality or hyperprecociality (*Hechenleitner, Grellet-Tinner & Fiorelli, 2015*; *Curry Rogers et al., 2016*). Precociality requires a relatively greater amount of available nutrients and therefore a larger egg size. Egg internal diameter constitutes a valuable proxy for the size of a fully developed embryo, so its precise measurement is important to figure out how big (and, eventually estimate, how strong) the embryo could have been. The new data shows that the Sanagasta and Tama eggs have nearly the same internal space for accommodating an embryo. This suggests that the hatchlings of Sanagasta could have been strong enough to pip through (at least) a 1.5 mm thick eggshell (Table 1). However, hatching through a 7.9 mm thick shell, more than three times thicker than other titanosaur eggs (depending on which species), seems unlikely.

The characteristics present in the archosaur eggshells result from a compromise between several factors (*Board, 1982*). They must be strong enough to prevent fracture, but sufficiently weak to allow hatching. This relationship is corroborated by the statistical analysis of the present data, which shows an association between the eggshell thickness and strength of the eggs ($F_{(2,7)} = 107.1$, $R^2 = 0.96$, $p = 8.53^{-5}$; Fig. 5B). The titanosaur eggs show, in general, a good fit to the statistical model (Fig. 5C). However, the Sanagasta

eggs with thick shell fall entirely outside these predictions. According to the FEA, they were 14–45 times stronger than any other titanosaur eggs that have nearly the same space for accommodating a late term embryo, such as those of Tama and Boseong. Thus, the Sanagasta embryos would have had to invest a considerable amount of energy to be able to hatch, if the eggs kept their thickness constant during the whole incubation.

Recapitulating on the adaptive advantage of such a thick shell for the Sanagasta specimens, two reasons that are not mutually exclusive can be considered: mechanical strength and resistance to chemical abrasion. Most titanosaurs laid biologically and mechanically viable eggs with thinner shells (e.g., Auca Mahuevo, Toteşti), which rarely exceed 2 mm, thus suggesting that strength was not a primary reason for developing thick eggshells. This shows that the excessive thickness of the Sanagasta shells would not respond to a mechanical need (e.g., withstand shock from outside).

However, keeping the shells thick during the whole incubation process could have had serious consequences for the Sanagasta titanosaurs. First, it would be detrimental for the development of the embryo because, as it grows, its needs change from preventing water loss to increasing gas exchange, due to the increase in energy consumption of a late embryo (a process documented among mound-nester archosaurs (*Ferguson, 1981*; *Booth & Seymour, 1987*; *Hechenleitner et al., 2016a*)). Second, a very thick eggshell might also represent a problem during hatching, as is suggested by the new results (Figs. 2A and 3). The case was pointed out by empirically studying *Alligator mississippiensis*, which bury their eggs in mounds of vegetation, in a way similar to that used by some titanosaurs and megapode birds (*Hechenleitner, Grellet-Tinner & Fiorelli, 2015*). Eggs incubated artificially (without natural substrate) develop normally, but then, the fully grown embryos are unable to break their shell (*Ferguson, 1981*). In nature, the dissolution of the *Alligator mississippiensis* eggshell is mediated by bacterial decomposition, which acidifies the nesting environment. Given the environmental similarities for ground-nesting, it is not surprising that the shells of several titanosaur nesting sites show evidence of extrinsic dissolution (*Grellet-Tinner, Chiappe & Coria, 2004*; *Hechenleitner, Grellet-Tinner & Fiorelli, 2015*). This type of dissolution should not be confused with the internal calcium absorption produced in the late stages of the embryogenesis, which is ubiquitous among archosaurs (*Chien, Hincke & McKee, 2009*). During ossification the calcium is removed from the shell, getting to reduce up to 20% of its thickness in precocial birds, such as the megapodes (*Booth & Seymour, 1987*). However, these high values are associated with very thin eggshells, in which the removal mostly affects the base of the structural units of calcite, in the innermost portion of the shell. Indeed, some internal dissolution in the Sanagasta eggshells was related with calcium resorption, but is negligible compared to the shell's thickness (*Grellet-Tinner, Fiorelli & Salvador, 2012*).

The results of FEA conducted on models of Sanagasta eggs with different shell thicknesses, between the minimum and maximum shell thickness reported for this site, show that an effort similar to the one necessary to break other titanosaur eggs would have had very little effect on those of Sanagasta immediately after oviposition (Figs. 4A and 4B). However, when the thickness is reduced to less than 1.6 mm, the shell becomes as fragile as for other titanosaur eggs.

The nesting strategies of titanosaurs have been compared with those of modern megapodes (*Kerourio, 1981*; *Cousin & Breton, 2000*; *Garcia et al., 2008*; *Grellet-Tinner & Fiorelli, 2010*; *Grellet-Tinner, Fiorelli & Salvador, 2012*; *Hechenleitner, Grellet-Tinner & Fiorelli, 2015*; *Grellet-Tinner, Lindsay & Thompson, 2017*). To date, only a handful of dinosaur species are confirmed to exploit and have exploited the geothermalism as a source of heat for incubating their eggs (*Jones & Birks, 1992*; *Grellet-Tinner & Fiorelli, 2010*; *Harris, Birks & Leaché, 2014*; *Hechenleitner, Grellet-Tinner & Fiorelli, 2015*; *Grellet-Tinner, Lindsay & Thompson, 2017*). The eggshell structure of modern dinosaurs differ from those of their ancestors by having three to four structural layers that confer a greater strength for a thinner eggshell thickness (*Grellet-Tinner, 2006*), instead of one structural layer like the Sanagasta dinosaur eggs. *Macrocephalon maleo* and *Megapodius pritchardii* are two modern megapode species that resort or revert to geothermal incubation, although the former, in Sulawesi Island, have two populations that do not interbreed and respectively utilize black sand with solar radiation and geothermal heated sand. However, the latter do oviposit in sands heated by in geothermal activities and *Megapodius pritchardii* in the volcanic ashes of calderas. In both instances the megapode eggs are not in direct contact with geothermal fluids. *Leipoa ocellata* and *Alectura lathami*, two mound-builder megapodes that inhabit Australia, must also deal with the risks of external acidic erosion. In their mound-nests the activity of microorganisms that maintains a high incubation temperature (*Seymour & Ackerman, 1980*) also produces organic acids as a by-product (*Grellet-Tinner, Lindsay & Thompson, 2017*). The eggshells of both species have an accessory layer composed of nanospheres of calcium phosphate on their outer surface (*Board, 1980*). *D'Alba et al. (2014)* showed that this accessory layer has antimicrobial properties. In addition, the calcium phosphate of the nanospheres is, compared to the calcite present in the structural layers of the eggshell, a relatively insoluble salt (*Board, 1980*). For this reason it has been recently suggested that the accessory layer also constitutes a protective cover that prevents the external erosion of the shell (*Grellet-Tinner, Lindsay & Thompson, 2017*). In addition, the pronounced nodular surficial ornamentation of these eggs complements the calcium phosphate nanospheres against chemical erosion by limiting most of the external erosion of their eggshell to these nodes. Therefore, although a few species of modern megapodes may display a reversal that utilizes ground generated heat as a passive incubating energy, their incubating strategies differ from the Sanagasta dinosaurs, which eggs were in direct contact with acidic geothermal fluids (*Grellet-Tinner & Fiorelli, 2010*).

## CONCLUSIONS

The FEA data suggest that hatching through a 7.9 mm thick shell was impossible for the embryos from Sanagasta. However, the analyses carried out on egg models with different shell thicknesses further suggest that thinning below 2 mm would have allowed these titanosaurs to hatch. With regard to the relationship between eggshell thickness and egg strength, the thick-shelled Sanagasta eggs are completely out of the prediction of the statistical model. In other words, the model shows that in terms of the strength/thickness ratio, the Sanagasta eggshells are disproportionately thick with respect to

those recorded for birds and other titanosaurs. As the original thickness would have been a strong limitation for hatching, the present results are consistent with previous arguments of outer eggshell thinning in the Sanagasta nesting site (*Grellet-Tinner & Fiorelli, 2010*; *Grellet-Tinner, Fiorelli & Salvador, 2012*). Considering that titanosaur eggs were incubated in fairly acid nesting environments, such as mounds or dug-out holes as seen in the modern megapodes (*Hechenleitner, Grellet-Tinner & Fiorelli, 2015*), it is plausible that the force required for hatching would be even less than estimated. Regardless of the factors (intrinsic and/or extrinsic) involved in the wear of ~80% of the eggshell, our results strongly suggest that external chemical dissolution, here complemented by the typical internal ontogenetic dissolution, throughout the incubation process would have been essential for allowing hatching of the titanosaurs that nested at Sanagasta.

## ACKNOWLEDGEMENTS

We thank the Secretaría de Cultura and Gobierno de La Rioja, Municipalidad de Tama and Sanagasta for their help and support. We also thank Alfredo Sangiorgio and the Hospital de la Madre y el Niño, La Rioja, for the access to the CT equipment.

### Funding

This work was supported by the Jurassic Foundation (2015) and PUE 2016 CONICET-CICTERRA. The funders had no role in study design, data collection and analysis, decision to publish, or preparation of the manuscript.

### Grant Disclosures

The following grant information was disclosed by the authors:
PUE 2016 CONICET-CICTERRA.
Jurassic Foundation (2015).

### Competing Interests

The authors declare that they have no competing interests.

### Author Contributions

- E. Martín Hechenleitner conceived and designed the experiments, analyzed the data, contributed reagents/materials/analysis tools, prepared figures and/or tables, authored or reviewed drafts of the paper, approved the final draft.
- Jeremías R. A. Taborda conceived and designed the experiments, performed the experiments, analyzed the data, contributed reagents/materials/analysis tools, prepared figures and/or tables, authored or reviewed drafts of the paper, approved the final draft.
- Lucas E. Fiorelli contributed reagents/materials/analysis tools, prepared figures and/or tables, approved the final draft.
- Gerald Grellet-Tinner authored or reviewed drafts of the paper, approved the final draft.
- Segundo R. Nuñez-Campero analyzed the data, approved the final draft.

## Data Availability

The configuration files for Finite Element Analyses are provided in the Supplemental File.

Hechenleitner, E. Martín; Taborda, Jeremías (2018): CRILAR-Pv 400 SA-C6-e1.rar. figshare. Figure. https://doi.org/10.6084/m9.figshare.6269879.v1.

## Supplemental Information

Supplemental information for this article can be found online at http://dx.doi.org/10.7717/peerj.4971#supplemental-information.

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
