# Peer review of "Biomechanical evidence suggests extensive eggshell thinning during incubation in the Sanagasta titanosaur dinosaurs"

_PeerJ, doi:10.7717/peerj.4971_

## Round 0.1 · original submission · Minor Revisions

The reviewers provide a number of helpful suggestions that should be considered during revision. In particular, I wish to highlight:

1) Move the supplemental Excel file into a main text table; it is small enough and important enough for the methods that it should be more than a supplement.

2) As recognized by the reviewers, much more methodological information should be provided. E.g., what types of elements and what types of meshes were used in the FEM?

3) One reviewer noted some missing citations and ootaxonomic updates that should be incorporated.

4) Because CT scan data are an important part of the project, please give strong consideration to formally archiving them in an appropriate institutional or community repository.

These and other suggestions should be addressed in your revision and rebuttal letter.

·

Basic reporting

see General comments for the author

Experimental design

see General comments for the author

Validity of the findings

see General comments for the author

Additional comments

Review of “Biomechanics suggests extensive eggshell thinning during incubation in a titanosaur dinosaur” by Hechenleitner et al.

This paper suggests that eggshells of titanosaur dinosaurs would have been through the process of dissolution during incubation based on mechanical analyses of the unusually thick eggshells. The main idea is clear and makes good sense. The paper is, in general, well-written, and the results are interesting. However, the following questions and concerns should be addressed before it may be considered for publication in PeerJ.

1. The main results of this paper are based on biomechanics; more details about the mechanical analyses should be provided. For example, what element types and meshes were used in the finite element analysis? How was the angle for the boundary conditions determined? An important assumption used in the present study is that the eggshell breaks when the displacement at the load point equals the thickness. Juang et al. reported in a recent paper that the eggshells of many avian species fractured at a displacement to thickness ratio of about 0.3 (see Fig. 5(b) in Juang et al “The avian egg exhibits general allometric invariances in mechanical design," Scientific Reports, 7, 14205, Oct. 2017). This ratio varies with species and almost all eggshells fractured when it reaches one. Although the actual ratio may vary with species, I agree that a ratio = 1 may be used as a simplified criteria to determine the fracture force. However, I suggest the authors to cite the above-mentioned reference and to elaborate a little more on this.

2. Some megapodes species, such as Macrocephalon maleo, also use burrow nesting with vegetal decomposition, solar radiation and/or geothermalism as incubating strategies. (see the corresponding author’ earlier work PeerJ 3:e1341, 2015). However, Maleo’s eggshell thickness is very thin, compared to that of other species with similar egg weight (Maleo: thickness = 0.38 mm, egg weight = 222 g, and Grus Antigone: thickness = 0.61 mm, egg weight = 217 g, see Juang et al. Scientific Reports, 7, 14205, Oct. 2017). If thick shell was an adaptation to geothermal environment, why did not Megapodes evolve thicker shells similar to the Sanagasta eggs, but, on the contrary, evolve a much thinner shell even thinner than “normal” avian ones?

3. What does “The new results” in line 169 mean? How do they reinforce the idea that the eggs of all titanosaurs were spherical? The connection between the current research and the reference studies should be elaborated more for the reader to understand what the authors are trying to convey from the text itself.

4. Please consider improving some phrasing for readers to comprehend more easily. For example, lines 40-42, 68-71, 74-76, 109-112, and 225-226. Also, more description should be provided for Figure 3. What does the numbering of the red dots mean? The explanation for Figure 3 in the Results section is not easy to understand, either. The data sources and the number of samples should be included in Table 1.

·

Basic reporting

The present manuscript (Ms. Ref. No. #24881) presents the results from investigations of the Biomechanics suggests extensive eggshell thinning during incubation in a titanosaur dinosaur. The article by Martín Hechenleitner et al. on biomechanics of a titanosaur dinosaur is well constructed and well presented. The wording and phrasing of the whole text is well written. The title of the paper is very informative.
All of the figures are relevant and are of high quality, well labelled and described. The research question is well defined and is pertinent and important. This research work fills the knowledge gap that how the thinning below 2 mm would have allowed these titanosaurs to hatch.

Experimental design

Normally the thickness of the titanosurid eggshells varies from 1.00-4.8 mm but the present record of thick eggshells from Sangasta locality of about 7.9 mm is quiet interesting and certainly helps for the nesting micro environments. The main aim of this manuscript is to test the mechanical implications of the thickest egg shells and how it pretentious the success of hatching. The authors have mentioned about the locality Auca Mahuevo in the abstract and text but not have clearly mentioned about the relation of these titanosaur eggs with newly erected Fusioolithus baghensis (Fernandez and Khosla, 2015) known from Late Cretaceous deposits of Argentina and India. This is the only titanosaurid egg which has the confirmed the presence of embryonic remains in the eggs from Argentina (Chiappe et al., 1998). In size, shape, micro and ultrastructiral studies indicates similarity between Late Cretaceous eggs of India and Argentina. The introduction part needs to define in a more detail manner. The authors should write in detail some of the earlier important works of Simon (2006) who erected the oospecies Patagoolithus salitrlensis and M. patagonicus (Calvo et al., 2009). The work of Khosla and Sahni (1995); Vianey-Liaud et al. (2003); Salgado et al. (2007, 2009) and Fernandez (2013, 2016) is completely missing, which needs to be added. I suggest that authors should start and improve the introduction description at lines 33- 42 to provide more justification for your study (specifically, you should expand upon the knowledge gap being filled). It is no where mentioned about the recent work on Indian and Argentinean dinosaur eggshells. The authors should also add that to date Argentinean titanosaurid eggshell parataxonomy has been revised by Fernandez and Khosla (2015) and has been restricted to five oospecies (Megaloolthus cylindricus, M. Jabalpurensis, M. megadermus, Fusioolithus baghensis and F. berthei, Fernandez and Khosla, 2015). The English language of at least two paragraphs should be enhanced to make certain that an international audience can obviously appreciate the present text. Some instance where the language could be enhanced includes lines 130-144. The current phrasing makes understanding complicated. The authors should add about how many number of samples they used for the present study in the methodology part. In order to make the manuscript more strong add the following references:
Ferna´ndez MS. 2013. Ana´lisis de ca´scaras de huevos de dinosaurios de la Formacio´n Allen, Creta´cico Superior de Rı´o Negro (Campaniano- Maastrichtiano): utilidades de los macrocaracteres de intere´s parataxono´mico. Ameghiniana. 50(1):79–97.
Fernandez and Khosla, A. 2015. Parataxonomic review of the Upper Cretaceous dinosaur eggshells belonging to the oofamily Megaloolithidae from India and Argentina. Historical Biology, 2015, Vol. 27, No. 2, 158–180.
Khosla A, Sahni A. 1995. Parataxonomic classification of Late Cretaceous dinosaur eggshells from India. J Paleontol Soc India. 40:87–102.
Salgado L, Coria RA, Ribeiro CM, Garrido A, Rogers R, Simo´n ME, Arcucci AB, Rogers KC, Carabajal AP, Apesteguı´a A, et al. 2007. Upper Cretaceous dinosaur nesting sites of Rı´o Negro (Salitral Ojo de Agua and Salinas de Trapalco´-Salitral de Santa Rosa), northern Patagonia, Argentina. Cretaceous Res. 28:392–404.
Salgado S, Ribeiro CM, Garcı´a RA, Ferna´ndez MS. 2009. Late Cretaceous Megaloolithid eggs from Salitral de Santa Rosa (Rı´o
Negro, Patagonia, Argentina): inferences on the titanosaurian reproductive biology. Ameghiniana. 46(4):605–620.
Simo´n ME. 2006. Ca´scaras de huevos de dinosaurios de la Formacio´n Allen (Campaniano-Maastrichtiano), en Salitral Moreno, provincia de Rı´o Negro, Argentina. Ameghiniana. 43(3):513–518.
Vianey-Liaud M, Khosla A, Garcı´a G. 2003. Relationships between European and Indian dinosaur egg and eggshells of the oofamily Megaloolithidae. J Vertebr Paleontol. 23(3):575–585.

Validity of the findings

The data provided in eggshell mechanical properties, egg morphology, statistical analysis and results are excellent. I suggest the authors to expand slightly the text of statistical part. It would be nice if the authors add more about hatching, paleoenvironments of these thick eggshells from Sanagasta. A separate paragraph about the paleoenvironmental and palaeoecological implications in the discussion part would make this manuscript strong and will be much appreciated by the international audience. In lines 235-237 the reference of Garcia et al. (2008, Paleovertebrata) should be added because they made comparative study between Megapodes and titanosaur eggs and their hatching patterns. Overall the data used in this manuscript is vigorous, statistically sound, and controlled. The conclusions are well stated, linked to original research question and should add more about palaeoenvironments and comparative hatching pattern of Megapodes and titanosaur eggs.

Additional comments

The present manuscript is well constructed, well presented and well written. The title of the paper is very informative. All of the figures are relevant and are of high quality, well labelled and described. This seems to be a unique work on the biomechanics of the titanosaurid eggshells. I strongly recommend that the manuscript should be accepted after minor revision.

Reviewer 3 ·

Basic reporting

This is a relatively short paper that purports to indicate that the particularly thick sauropod eggshells reported from one location are due to the nesting environment. They support a previous paper by using primarily FEA to suggest the eggshell was initially very thick and then etched by environmental acid to a thickness where the hatchling could hatch. I am not sure that there is much that is that novel in the study. This impression is reinforced by the lack of any real detailed description of the FEA methodology and there seems to be just an extrapolation of data from Hahn et al. to the larger dinosaur eggs.

Much more is needed to explain the methodology here to provide a sound context for the results.

Experimental design

The data seem to be based on single specimens (often a consequence of dealing with fossils) or from existing published data. The methodology was based primarily based on modelling the characteristics of the outline shape of the egg.

Given that this is an on-line journal it is unclear why any supplementary materials is required - surely methods and results could be included in the main paper.

Validity of the findings

Basically, the conclusion is that sauropod eggs were spherical - already well known and that the thick-shelled examples would have been etched by an acidic environment to a thickness that would allow a hatchling to escape. This idea has already been published and the problem I have with the paper is that there is a lack of clarity and depth as to why this particular study is any different from previous studies.

The authors need to better address the topics concerned and explain how their short paper adds to our understanding of the topic.

---

## Round 0.2 · Minor Revisions

Thank you for your careful attention to the comments from the reviewers. The manuscript is much improved, and in my view it is nearly ready for publication. I have made one last pass through the file, and added some minor style/grammar/spelling corrections, as well as a few other notations.

In your resubmission, you had a note that the CT data are being archived. Please include the repository information with your resubmission.

---

## Round 0.3 · accepted · Accept

Thank you for your attention to the most recent round of comments.

#